# Enhancing Agentic textual Graph Retrieval with Synthetic Stepwise Supervision

## Abstract

A significant portion of real-world data is inherently represented as textual graphs, and integrating these graphs into large language models (LLMs) is promising to enable complex graph-based question answering. However, a key challenge in LLM-based textual graph QA systems lies in graph retrieval, i.e., how to retrieve relevant content from large graphs that is sufficiently informative while remaining compact for the LLM context. Existing retrievers suffer from poor performance since they either rely on shallow embedding similarity or employ interactive retrieving policies that demand excessive data labeling and training cost. To address these issues, we present *Graph-S*$^3$, an agentic textual graph reasoning framework that employs an LLM-based retriever trained with synthetic stepwise supervision. Instead of rewarding the agent based on the final answers, which may lead to sparse and unstable training signals, we propose to closely evaluate each step of the retriever based on offline-extracted golden subgraphs. Our main techniques include a data synthesis pipeline to extract the golden subgraphs for reward generation and a two-stage training scheme to learn the interactive graph exploration policy based on the synthesized rewards. Based on extensive experiments on three common datasets in comparison with seven strong baselines, our approach achieves an average improvement of 8.1% in accuracy and 9.7% in $F_1$ score. The advantage is even higher in more complicated multi-hop reasoning tasks. Our code will be open-sourced.

## 1 Introduction

Textual graphs are graphs with text-attributed nodes and edges, which are widely used for structured knowledge representation with many applications in question answering, recommendation, and scientific discovery (Peng et al., 2024; Procko & Ochoa, 2024; Zhang et al., 2025a). By explicitly modeling multi-hop relations and semantic constraints, textual graphs enable interpretable and compositional reasoning that is difficult to achieve with unstructured text corpora (Chen et al., 2020; Hogan et al., 2021; Zou, 2020).

The early approaches to reasoning over textual graphs relied on costly annotations and inflexible symbolic inference (Yih et al., 2016). The emergence of large language models (LLMs) has alleviated these limitations through strong semantic understanding (Chang et al., 2024), inspiring a growing body of work that combines LLMs with textual graphs for general-purpose graph understanding and question answering (QA) (Lewis et al., 2020; Peng et al., 2024; Procko & Ochoa, 2024; Zhang et al., 2025a; Jin et al., 2024; Chai et al., 2023).

A key step in LLM-based textual graph QA systems is graph retrieval, i.e., retrieving the relevant content from the target graph based on the natural language query. Most existing retrievers encode graph nodes and edges as vector embeddings and perform similarity-based matching with the query (Chen et al., 2024b;a; Tang et al., 2024). This approach has the advantage of efficiency, but it often produces noisy or incomplete results due to the coarse-grained matching process. Moreover, these methods usually retrieve a large neighborhood in a single step and then flatten all candidate triples into text for the LLM (Edge et al., 2024; Guo et al., 2024b; He et al., 2024). Such flattening discards the relational structure of the graph and obscures reasoning trajectories, which degrades performance on long multi-hop reasoning tasks.

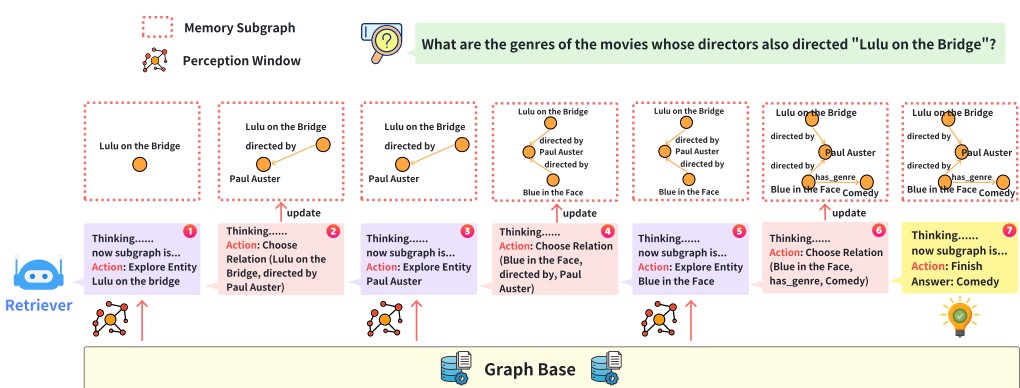

Figure 1: An illustration of agentic textual graph retrieval and question answering.

Another line of work employs LLM-based agents that access graph information through tool calls (Jiang et al., 2024; Yang et al., 2024; Ji et al., 2024). Although these methods have the potential to surpass simple similarity retrieval, their training is typically bootstrapped using supervised fine-tuning (SFT). This strategy causes the action space of the model to collapse (Chu et al., 2025; Li et al., 2024), leading it to memorize patterns from the training data rather than learning generalizable policies. As a result, such agents struggle to locate optimal reasoning trajectories. Moreover, constructing these trajectories requires substantial expert annotations, making such approaches difficult to scale.

To this end, we propose *Graph-S³*, an agentic retrieval framework that equips an LLM-based retriever with the ability to perform interactive, structure-aware exploration of textual graphs. Figure 1 provides an example of agentic retrieval, showing how the retriever agent iteratively performs actions to collect the necessary information to solve the given query.

To train such an LLM-based agent system, a straightforward approach is to employ outcome-supervised reinforcement learning (Lightman et al., 2023; Paolo et al., 2024), i.e., rolling out lots of reasoning trajectories in the graphs with a policy and optimize the policy with reward signals computed from the final answers. However, this approach is difficult to apply in practice for textual graph reasoning, since the outcome-based rewards are often sparse and unstable: The action space for real-world textual graphs is often too large to efficiently discover the optimal retrieval path, while redundant or erroneous retrieval steps may still lead to the correct final answer, making outcome supervision an unreliable signal of reasoning quality (Liu et al., 2023; Rengarajan et al., 2022). This challenge hinders the direct adoption of standard RL algorithms to the problem of textual graph reasoning.

To overcome this limitation, we introduce a synthetic stepwise supervision scheme that provides explicit feedback at every decision step, ensuring that the model is guided not only by the correctness of the final answer but also by the quality of intermediate actions. The key idea is to **guide each graph retrieval step with golden subgraphs offline extracted from the target graph**. Specifically, we propose an automated pipeline to construct the golden subgraphs for reward computation. We first generate a large amount of subgraph candidates through random and LLM-guided exploration, and filter the candidates based on **information sufficiency**, i.e. whether they are able to produce correct answers with LLMs. These successful exploratory trajectories are used for SFT, providing the retriever with basic navigation ability as a warm-up stage. Then we further refine the subgraphs to enhance **information conciseness** by iteratively pruning redundant content while preserving answer consistency. With these refined subgraphs, each online graph retrieval action can be associated with an explicit stepwise reward based on its contribution to the golden subgraphs. The combined two-step training pipeline guides the retriever to improve reasoning decisions over long action chains.

Extensive experiments have demonstrated the effectiveness of our synthetic stepwise supervision approach. For example, while retrieving only 11.44% of the triples, *Graph-S³* achieves an average

improvement of 8.1% in accuracy and 9.7% in $F_1$ score across the WebQSP, CWQ, and MetaQA benchmarks.

In summary, the main contributions of this work are as follows:

(1) We propose an automatic pipeline for synthesizing high-quality stepwise supervision data for interactive graph retrieval, addressing the scarcity of fine-grained training signals in this field.

(2) We design a two-stage training paradigm tailored for graph reasoning: supervised fine-tuning on raw synthetic trajectories to bootstrap basic navigation ability, followed by reinforcement learning with synthetic stepwise rewards on refined trajectories to provide explicit feedback and strengthen reasoning strategies.

(3) Experimental results demonstrate that *Graph-S$^3$* achieves state-of-the-art performance on the WebQSP, CWQ, and MetaQA datasets with accurate and compact graph retrieval.

## 2 RELATED WORK

### 2.1 GRAPH RETRIEVAL METHODS

Graph retrieval approaches include similarity-based, GNN-based, and LLM-based methods (Peng et al., 2024; Procko & Ochoa, 2024; Zhang et al., 2025a; Zhu et al., 2025; Han et al., 2024), but most perform one-shot retrieval and often return redundant or incomplete subgraphs. Recent interactive frameworks (Jiang et al., 2024; Ji et al., 2024; Yang et al., 2024) allow iterative exploration, yet their training predominantly relies on imitation of language patterns or outcome-based supervision, which provides only coarse feedback and limits stable multi-hop reasoning. In contrast, our work employs reinforcement learning with synthetic stepwise rewards and a scalable data synthesis pipeline to provide supervision for interactive retrieval.

### 2.2 STEPWISE REINFORCEMENT LEARNING FOR GRAPH REASONING

Recent advances such as OpenAI o1 and DeepSeek-R1 (Jaech et al., 2024; Guo et al., 2025) demonstrate the effectiveness of reinforcement learning in strengthening multi-step reasoning, enabling models to perform longer chains of thought with improved reliability in domains like mathematics and programming (Guo et al., 2024a; El-Kishky et al., 2025). In contrast, applying RL to textual graphs remains limited, partly due to the lack of fine-grained supervision data (Zhang et al., 2025a; Yao et al., 2025; Liu et al., 2025). RL-based graph agents (Das et al., 2017; Cui et al., 2025) relied on sparse outcome rewards, making credit assignment across reasoning trajectories difficult. Although recent efforts have introduced reasoning-structured datasets (Pahilajani et al., 2024), high-quality stepwise supervision for graph-based RL remains scarce and difficult to construct at scale. These observations highlight the need for scalable approaches that can provide fine-grained supervision signals and support stable optimization for interactive graph retrieval.

## 3 METHOD

We present our agentic retrieval framework, designed to equip large language models with robust graph reasoning capabilities through stepwise supervision, a two-stage training paradigm, and an interactive retrieval strategy. As illustrated in Figure 2, the framework comprises three main components. First, we construct an automatic data synthesis pipeline that leverages GPT-4o (OpenAI, 2024) to generate diverse exploratory trajectories, which are subsequently refined into high-quality stepwise supervision data. This addresses the scarcity of fine-grained training signals for graph-based reinforcement learning. Second, we adopt a two-stage training paradigm: SFT on raw synthetic trajectories provides a warm-up initialization for basic graph navigation, while RL with synthetic stepwise rewards on refined trajectories supplies explicit feedback at each decision step, stabilizing optimization and strengthening reasoning strategies. Finally, during inference, the retriever operates under an interactive retrieval mechanism that conducts stepwise, structure-aware exploration of the textual graph, thereby reducing redundancy and mitigating incomplete retrieval.

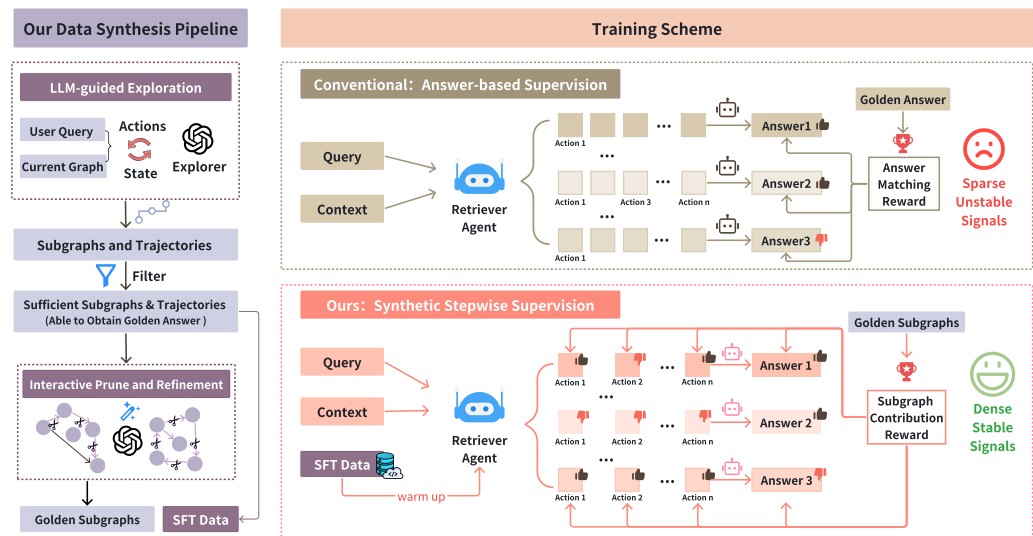

Figure 2: An Overview of our data synthesis pipeline and training scheme.

## 3.1 DATA SYNTHESIS

Existing LLMs are not pretrained on graph-structured data (Zhang et al., 2025b), which significantly limits their performance on graph reasoning tasks. As a result, effective training requires high-quality supervision to cultivate graph comprehension and reasoning capabilities. However, constructing such datasets is notoriously expensive, since it often relies on manual annotation by domain experts (Choubey et al., 2024), leading to a persistent scarcity of high-quality graph reasoning data. To address this issue, we design a pipeline that automatically synthesizes graph reasoning trajectories. Specifically, we first define a set of actions that enable structured interaction with the graph. Then, we leverage GPT-4o to perform these actions and collect valid action–response pairs, which form exploratory trajectories. The raw trajectories are directly used for SFT to provide the model with basic navigation ability, while a refinement step prunes redundant detours and preserves all answer-consistent trajectories, producing high-quality stepwise supervision data for RL.

### 3.1.1 ACTION SPACE FOR GRAPH EXPLORATION

Given a textual graph $\mathcal{G} = \{t_i\}_{i=1}^m$, where each triple $t_i = (e_h^i, r^i, e_t^i) \in \mathcal{E} \times \mathcal{R} \times \mathcal{E}$ consists of a head entity $e_h^i \in \mathcal{E}$, a relation $r^i \in \mathcal{R}$, and a tail entity $e_t^i \in \mathcal{E}$. Here, $\mathcal{E}$ and $\mathcal{R}$ denote the sets of entities and relations, respectively. To enable stepwise exploration over $\mathcal{G}$, we define the retriever's action space as consisting of three types of operations. For clarity, we use $(x, r, y)$ to denote a generic triple in $\mathcal{G}$, where $x, y \in \mathcal{E}$ and $r \in \mathcal{R}$.

**Explore Entity**: This operation expands the local neighborhood of a given entity by retrieving all directly connected triples in $\mathcal{G}$. Formally, for a target entity $x \in \mathcal{E}$, the operation is defined as

$$\text{Explore}(x) = \{(x, r, y) \mid (x, r, y) \in \mathcal{G}\} \cup \{(y, r, x) \mid (y, r, x) \in \mathcal{G}\}, \tag{1}$$

where $r \in \mathcal{R}$ and $y \in \mathcal{E}$ denote relations and neighboring entities, respectively. The retrieved triples are added to the perception window $\mathcal{G}^p$ for subsequent reasoning steps.

**Choose Relation**: The perception window $\mathcal{G}^p$ obtained from the EXPLORE action may still contain many irrelevant triples. To avoid introducing redundant context into the LLM, this operation prunes $\mathcal{G}^p$ into a query-relevant subgraph $\mathcal{G}^{sub}$:

$$\text{Choose}(q, \mathcal{G}^p) = \{(x, r, y) \in \mathcal{G}^p \mid F(q, (x, r, y)) = 1\}. \tag{2}$$

Here, $q$ is the query, $(x, r, y)$ denotes a triple in $\mathcal{G}^p$, and $F(q, (x, r, y))$ is the relevance function learned by the retriever to decide whether the triple should be preserved. $F$ outputs 1 if the triple is judged relevant to the query and 0 otherwise.

**Finish**: This operation indicates that the retriever has gathered sufficient evidence in the current sub-graph $\mathcal{G}^{sub}$ to answer the query $q$. Once invoked, the exploration process terminates, and $(q, \mathcal{G}^{sub})$ is used to produce the final answer:

$$\text{Finish}(q, \mathcal{G}^{sub}) = \text{Answer}(q, \mathcal{G}^{sub}), \tag{3}$$

where $\text{Answer}(q, \mathcal{G}^{sub})$ denotes answering query $q$ based on the retrieved subgraph.

### 3.1.2 GRAPH REASONING DATA SYNTHESIS

Given the defined action space, we synthesize reasoning trajectories by letting a behavior model (GPT-4o) interact with the graph. We formalize the generation process as a Markov decision process (MDP) with deterministic transitions defined by the graph action space. Each trajectory consists of multiple decision steps, and is later decomposed into step-level training instances for SFT and RL.

**State.** We define each state as a tuple of four components: $s_t = \left(q, \ \mathcal{G}_t^p, \ \mathcal{G}_t^{sub}, \ h_t\right)$, where $q$ is the query, $\mathcal{G}_t^p$ the perception window aggregated by EXPLORE, and $\mathcal{G}_t^{sub}$ the focused subgraph maintained by CHOOSE, and $h_t = (a_1, \ldots, a_{t-1})$ the action history up to step $t$. Including $h_t$ allows the retriever to condition its decisions not only on the current graph view but also on the reasoning trajectory already taken.

**Action.** The parameterized action space is $\mathcal{A}$, defined in 3.1.1. During the synthesis phase, the behavior model selects $a_t \in \mathcal{A}$ given $s_t$, and additionally produces a natural language reasoning trace that explains the choice of action.

**Transition.** Executing $a_t$ updates the state via graph action space:

$$s_{t+1} = \begin{cases} \left(q, \ \mathcal{G}_t^p \cup \text{EXPLORE}(x_t), \ \mathcal{G}_t^{sub}, \ h_t \cup \{a_t\}\right), & a_t = \text{EXPLORE}(x_t), \\ \left(q, \ \mathcal{G}_t^p, \ \{(x,r,y) \in \mathcal{G}_t^p \mid F_B(q, (x,r,y)) = 1\}, \ h_t \cup \{a_t\}\right), & a_t = \text{CHOOSE}, \\ \left(q, \ \mathcal{G}_t^p, \ \mathcal{G}_t^{sub}, \ h_t \cup \{a_t\}\right), & a_t = \text{FINISH}, \end{cases} \tag{4}$$

An episode terminates when $a_t = \text{FINISH}$ or when $t$ reaches a preset limit $T_{\max}$.

**Answer and retention.** Upon termination at step $T$, we produce an answer $\hat{y} = \text{Answer}\left(q, \mathcal{G}_T^{sub}\right)$ and keep the trajectory $\tau = \left\{(s_t, a_t)\right\}_{t=1}^T$ only if $\hat{y}$ matches the set of ground-truth answers. After that, we use the raw action labels for SFT dataset: $\mathcal{D}_{\text{SFT}} = \left\{(s_t, a_t)\right\}_{\tau, t}$. In the next subsection, we further refine trajectories to obtain stepwise supervision signals for RL.

### 3.1.3 TRAJECTORY REFINEMENT FOR REINFORCEMENT LEARNING

While supervised fine-tuning directly benefits from raw trajectories, reinforcement learning requires more concise training signals (Yue et al., 2025). Filtering trajectories only by final answer correctness often produces redundant exploration steps, introducing noise and inefficiency during policy optimization. To address this, we introduce a refinement procedure that removes unnecessary detours while preserving all answer-consistent trajectories, thereby yielding shorter and cleaner trajectories for RL.

Let the set of retained raw trajectories be $\mathcal{T} = \{\tau_i\}_{i=1}^N$, where each $\tau_i = \{(s_t, a_t)\}_{t=1}^{T_i}$ is a sequence of state–action pairs terminating at step $T_i$. We define a refinement operator $\mathcal{R}$ that maps the raw set $\mathcal{T}$ into a refined set $\mathcal{T}^*$:

$$\mathcal{T}^* = \mathcal{R}(\mathcal{T}). \tag{5}$$

For each $\tau_i$, the refinement identifies the shortest feasible subsequence $\tau_i^*$ that still leads to the same correct final answer:

$$\tau_i^* = \arg\min_{\tau \in \mathcal{F}_i} |\tau|, \tag{6}$$

where $\mathcal{F}_i$ is the set of all feasible answer-consistent trajectories equivalent to $\tau_i$:

$$\mathcal{F}_i = \{\tau \mid \text{FinalState}(\tau) = \text{FinalState}(\tau_i), \ \text{CorrectAnswer}(\tau) = \text{CorrectAnswer}(\tau_i)\}. \tag{7}$$

Thus, the refined dataset $\mathcal{T}^*$ retains trajectories that are semantically equivalent to the originals but stripped of redundant exploration steps. Each refined trajectory $\tau_i^*$ is then decomposed into

step-level supervision signals by attaching a rule-based stepwise reward $\ell_t \in [0, 1]$ to each action $a_t$, indicating its correctness within the reasoning trajectory. Formally, the RL training dataset is constructed as $\mathcal{D}_{\text{RL}} = \big\{(s_t, a_t, \ell_t)\big\}_{\tau^*, t}$. This ensures that reinforcement learning receives concise stepwise supervision signals, improving both stability and efficiency of training.

## 3.2 Training Stage

To enhance the model's graph comprehension and reasoning capabilities, we adopt a two-stage fine-tuning approach. The first stage uses SFT with synthesized data to establish foundational abilities. The second stage employs GRPO with trajectory refinement, leveraging RL's proven effectiveness in enhancing reasoning and exploration efficiency (Yue et al., 2025).

### 3.2.1 Stage I: Supervised fine-tuning

For each step $t$, let $s_t = (q, \mathcal{G}_t^p, \mathcal{G}_t^{sub}, h_t)$ be the serialized state, and let $y_t = (y_t^1, \ldots, y_t^{L_t})$ be the target token sequence that concatenates the natural-language thought process and the action specification. Denote by $\mathcal{I}(s_t)$ the textual serialization of the state. The training loss of SFT is defined as

$$\mathcal{L}_{\text{SFT}}(\theta) = - \mathbb{E}_{(s_t, y_t) \sim \mathcal{D}_{\text{SFT}}} \left[ \sum_{l=1}^{L_t} \log \pi_\theta\big(y_t^l \,\big|\, \mathcal{I}(s_t),\, y_t^{<l}\big) \right], \tag{8}$$

where $\pi_\theta(y_t^l \mid \mathcal{I}(s_t), y_t^{<l})$ denotes the probability assigned by the model to the $l$-th token given the serialized state and the previously generated tokens.

### 3.2.2 Stage II: Reinforcement learning with stepwise rewards

For reward design, existing approaches predominantly rely on outcome-based reward signals, which have demonstrated remarkable effectiveness in domains such as mathematical reasoning and code generation. However, prior studies (Wang et al., 2025; Choudhury, 2025; Deng et al., 2024) have shown that in relatively complex scenarios such as graph retrieval, conventional outcome-based reward signals tend to be overly sparse. This sparsity hampers effective credit assignment to early-stage actions, ultimately resulting in inefficient learning over long action chains. This observation motivates our adoption of process-level rewards in the training process, where the reward signal is determined by the contribution of each current action to the golden subgraphs.

Specifically, each step $t$ is associated with a process-level rule-based reward $\ell_t$ that provides graded feedback according to the quality of the predicted action:

$$\ell_t = \begin{cases} 0, & \text{if Invalid}(a_t), \\ c_1, & \text{if Format}(a_t) = 1 \,\wedge\, \text{ActionCorrect}(a_t) = 0, \\ c_2, & \text{if Partial}(a_t) = 1, \\ 1.0, & \text{if } a_t = a_t^*. \end{cases} \tag{9}$$

Here $\text{Invalid}(\cdot)$, $\text{Format}(\cdot)$, and $\text{Partial}(\cdot)$ are deterministic rule-based functions that check output validity, format correctness, and partial correctness, respectively, and $c_1, c_2$ are dataset-specific hyperparameters. Consequently, we adopt the reward function shown in Eq. 9 to train our model using the GRPO method.

## 3.3 Interactive Retriever

At inference time, *Graph-S$^3$* interacts with the textual graph through the defined action space. Unlike single-pass retrieval methods that return large subgraphs, our approach performs stepwise exploration by balancing EXPLORE and CHOOSE actions, terminating with FINISH when sufficient evidence is gathered. This interactive process enables precise control over retrieval depth while minimizing redundancy, producing concise subgraphs for reasoning.

## 4 EXPERIMENTS

| Retriever + Generator | WebQSP | | CWQ | | MetaQA 1-hop | | MetaQA 2-hop | | MetaQA 3-hop | |
|---|---|---|---|---|---|---|---|---|---|---|
| | Acc | $F_1$ | Acc | $F_1$ | Acc | $F_1$ | Acc | $F_1$ | Acc | $F_1$ |
| No graph + Qwen3-8B | 5.16 | 8.11 | 6.26 | 7.35 | 2.00 | 2.88 | 0.07 | 0.95 | 0.20 | 1.19 |
| No retriever + Qwen3-8B | 0.25 | 1.83 | 0.37 | 1.29 | 0.00 | 0.29 | 0.00 | 0.49 | 0.00 | 1.25 |
| RAG/1hop + Qwen3-8B | 27.89 | 38.57 | 12.55 | 16.18 | 75.93 | 86.36 | 0.77 | 1.74 | 4.13 | 11.99 |
| RAG/2hop + Qwen3-8B | 14.07 | 24.47 | 7.00 | 10.55 | 42.03 | 55.59 | 10.07 | 21.65 | 2.60 | 9.42 |
| RAG/3hop + Qwen3-8B | 1.54 | 7.94 | 0.99 | 3.12 | 0.37 | 3.03 | 0.13 | 2.21 | 0.13 | 3.45 |
| ToG + Qwen3-8B | 6.14 | 9.79 | 7.01 | 9.61 | 1.37 | 1.75 | 0.00 | 0.00 | 0.00 | 0.20 |
| LightRAG + Qwen3-8B | 18.39 | 31.67 | 16.20 | 23.09 | 1.13 | 1.76 | 0.00 | 0.19 | 0.07 | 0.40 |
| G-retriever + Qwen3-8B | 25.74 | 35.45 | 15.38 | 18.62 | 0.63 | 1.60 | 0.10 | 0.77 | 0.03 | 1.87 |
| KG-Agent + Qwen3-8B | 29.66 | 38.02 | 15.64 | 22.45 | 75.80 | 85.92 | 28.41 | 33.66 | 4.07 | 10.89 |
| *Graph-S$^3$* + Qwen3-8B | **36.24** | **47.88** | **17.87** | **23.29** | **81.50** | **90.22** | **53.73** | **65.60** | **12.73** | **29.49** |
| No graph + Llama3.1-8B | 8.97 | 15.69 | 9.40 | 11.58 | 12.20 | 17.60 | 1.27 | 7.77 | 1.23 | 8.86 |
| No retriever + Llama3.1-8B | 0.18 | 1.97 | 0.14 | 1.29 | 0.00 | 0.58 | 0.00 | 1.11 | 0.00 | 2.94 |
| RAG/1hop + Llama3.1-8B | 24.82 | 35.28 | 13.85 | 17.26 | 60.17 | 70.84 | 2.40 | 5.98 | 4.03 | 15.46 |
| RAG/2hop + Llama3.1-8B | 11.06 | 22.94 | 6.29 | 10.68 | 29.07 | 42.47 | 4.50 | 15.06 | 1.80 | 11.30 |
| RAG/3hop + Llama3.1-8B | 1.04 | 6.67 | 0.65 | 3.43 | 0.33 | 3.67 | 0.17 | 3.37 | 0.07 | 5.76 |
| ToG + Llama3.1-8B | 8.85 | 14.28 | 8.42 | 12.33 | 12.40 | 15.88 | 0.00 | 0.63 | 1.43 | 6.10 |
| LightRAG + Llama3.1-8B | 15.85 | 36.66 | 8.33 | 15.01 | 13.13 | 21.47 | 0.93 | 4.33 | 1.00 | 6.38 |
| G-retriever + Llama3.1-8B | 22.67 | 32.26 | 13.91 | 17.47 | 0.67 | 1.56 | 0.10 | 0.83 | 0.10 | 1.77 |
| KG-Agent + Llama3.1-8B | 32.04 | 41.99 | 10.82 | 13.75 | 65.55 | 75.03 | 32.51 | 44.52 | 3.85 | 8.29 |
| *Graph-S$^3$* + Llama3.1-8B | **32.31** | **43.26** | **17.11** | **21.17** | **67.50** | **76.56** | **40.17** | **55.49** | **10.73** | **29.55** |
| No graph + Finetuned-8B | 9.21 | 14.88 | 10.17 | 12.31 | 1.63 | 2.49 | 0.43 | 2.64 | 0.33 | 4.60 |
| No retriever + Finetuned-8B | 0.37 | 2.26 | 0.68 | 1.76 | 0.00 | 0.29 | 0.00 | 0.34 | 0.00 | 1.05 |
| RAG/1hop + Finetuned-8B | 28.87 | 41.48 | 19.85 | 26.01 | 59.50 | 69.54 | 1.83 | 7.44 | 3.30 | 18.61 |
| RAG/2hop + Finetuned-8B | 14.93 | 27.76 | 8.89 | 14.56 | 35.83 | 52.03 | 7.70 | 21.21 | 3.07 | 14.04 |
| RAG/3hop + Finetuned-8B | 1.54 | 7.81 | 0.93 | 4.36 | 0.57 | 2.99 | 0.43 | 2.31 | 0.13 | 4.23 |
| ToG + Finetuned-8B | 5.04 | 9.43 | 7.54 | 9.79 | 2.23 | 3.44 | 0.00 | 0.12 | 0.10 | 2.80 |
| LightRAG + Finetuned-8B | 17.38 | 32.59 | 13.85 | 19.96 | 13.77 | 20.60 | 0.97 | 3.71 | 0.53 | 4.28 |
| G-retriever + Finetuned-8B | 30.34 | 43.49 | 22.68 | 28.38 | 8.93 | 11.51 | 2.33 | 4.80 | 0.40 | 3.31 |
| KG-Agent + Finetuned-8B | 42.60 | 55.38 | 13.77 | 16.29 | 75.99 | 86.02 | 30.19 | 43.95 | 2.64 | 7.20 |
| *Graph-S$^3$* + Finetuned-8B | **44.29** | **58.45** | **23.62** | **30.44** | **82.77** | **92.04** | **63.17** | **76.18** | **14.70** | **36.29** |

Table 1: Overall results on graph-based QA benchmarks. The best results are highlighted in bold and the second performance results are indicated by an underscore.

### 4.1 EXPERIMENTAL SETUP

**Datasets.** We evaluate *Graph-S$^3$* on three widely used textual graph QA benchmarks. WebQSP (Yih et al., 2015) consists of real-world questions annotated with SPARQL queries against Freebase, primarily involving one- or two-hop reasoning. CWQ (Talmor & Berant, 2018) extends WebQSP with more complex multi-hop questions, posing a greater challenge for long reasoning chains. MetaQA (Zhang et al., 2018) is a movie-domain benchmark containing 135k triples and 43k entities, designed to evaluate multi-hop reasoning in a closed domain. Following prior work (Chen et al., 2024b), we report accuracy (Acc) and $F_1$ score as evaluation metrics.

**Baselines.** To validate the effectiveness of our approach, we compare with several representative graph retrieval methods. We additionally evaluate the model's inherent graph understanding capability through two configurations: (1) the *no graph* setting, where the model processes the query without any graph input, and (2) the *no retriever* setting, where the model receives the entire graph structure directly as input. For traditional RAG, we implemented a multi-hop method where the model retrieves the most relevant graph nodes for the current query and then performs a k-hop expansion to collect information for answer generation. We further compare with representative graph retrievers, including Think-on-Graph (ToG) (Sun et al., 2023), LightRAG (Guo et al., 2024b), G-Retriever (He et al., 2024) and agentic graph retrieval system KG-Agent (Jiang et al., 2024).

**Implementation Details.** In our experiments, we primarily employed the Llama3.1-8B (Dubey et al., 2024) and Qwen3-8B (Yang et al., 2025) models. Our data synthesis pipeline produced 9,035 SFT and 3,504 RL training instances; with this data, the Qwen3-8B model was trained for 3 SFT and 15 RL epochs on 8 A100 GPUs, requiring 32 hours in total. Further training details are provided in the Appendix A.3.

## 4.2 MAIN RESULTS

The results of WebQSP, CWQ, and MetaQA are summarized in Table 1. In general, our framework achieves the best performance among all compared methods, demonstrating the effectiveness of combining two-stage training with interactive retrieval. Compared with the no-retriever configuration, where the entire graph is directly fed into the LLM, retrieval-based methods consistently achieve higher accuracy, confirming that selective subgraph retrieval is essential since full graphs exceed the effective processing capacity of LLMs. Relative to $k$-hop expansion approaches, *Graph-$S^3$* yields clear improvements, particularly on multi-hop benchmarks, showing that interactive retrieval can effectively filter relevant relations while avoiding redundant context. Furthermore, against training-free baselines such as ToG and LightRAG, our model delivers substantial gains, highlighting the importance of stepwise synthetic supervision and reinforcement learning in enhancing the reasoning ability of the retriever. Compared with trained retrievers such as G-Retriever and existing agentic graph retrieval frameworks like KG-Agent, our approach further improves multi-hop reasoning performance, indicating that the introduction of reinforcement learning and stepwise supervision enables the retriever to acquire stronger reasoning ability beyond representation learning.

## 4.3 IN-DEPTH ANALYSIS

| Methods | Dataset | | | | | | | | | |
| | WebQSP | | CWQ | | MetaQA | | | | | |
| | | | | | 1hop | | 2hop | | 3hop | |
| | Acc | $F_1$ | Acc | $F_1$ | Acc | $F_1$ | Acc | $F_1$ | Acc | $F_1$ |
|---|---|---|---|---|---|---|---|---|---|---|
| *Graph-$S^3$* | **44.29** | **58.45** | **23.62** | **30.44** | **82.77** | **92.04** | **63.17** | **76.18** | **14.70** | **36.29** |
| *Graph-$S^3$* **w/o SFT** | 31.64 | 44.41 | 7.74 | 8.77 | 81.27 | 89.38 | 46.30 | 54.12 | 2.07 | 4.98 |
| *Graph-$S^3$* **w/o RL** | 41.77 | 53.02 | 13.39 | 15.97 | 71.97 | 80.09 | 35.93 | 45.25 | 5.73 | 11.46 |
| *Graph-$S^3$* **w/o interactive** | 28.87 | 41.48 | 19.85 | 26.01 | 59.50 | 69.54 | 1.83 | 7.44 | 3.30 | 18.61 |
| *Graph-$S^3$* **w/o trajectory refinement** | 16.46 | 19.24 | 4.12 | 4.87 | 39.47 | 41.06 | 4.01 | 6.10 | 1.34 | 1.80 |

Table 2: Results of abalation studies.

### 4.3.1 ABLATION STUDY

Our framework consists of four key components: supervised fine-tuning (SFT), reinforcement learning (RL) with stepwise rewards, interactive retrieval at inference time, and trajectory refinement during data synthesis. To assess the contribution of each component, we remove one module at a time and evaluate the resulting performance degradation. The results are reported in Table 2.

**Ablation of SFT.** Removing the SFT stage leads to a clear drop in Accuracy and $F_1$ across all benchmarks. This confirms that SFT provides the retriever with essential navigation ability, compensating for the lack of graph-specific training during upstream pretraining and establishing a stable foundation for subsequent RL optimization.

**Ablation of RL.** Eliminating the RL stage results in consistent performance degradation, with particularly large declines on CWQ and MetaQA, which require longer reasoning chains. This demonstrates that RL with stepwise rewards substantially strengthens the retriever's reasoning capability, especially on complex multi-hop tasks.

**Ablation of interactive inference.** Disabling interactive retrieval causes significant performance drops on 2-hop and 3-hop questions, where results approach those of conventional $k$-hop RAG. This shows that interactive retrieval is crucial for adaptively controlling retrieval depth, effectively filtering redundant neighbors while preserving critical relations.

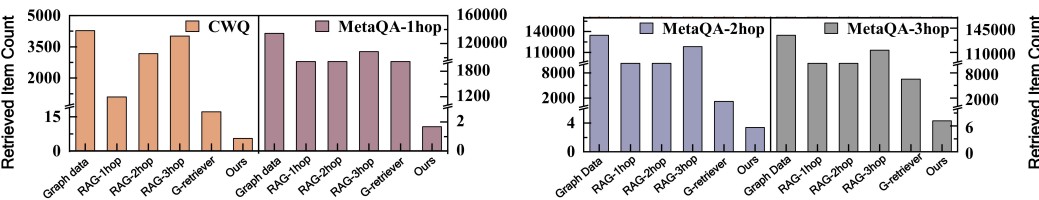

Figure 3: Number of retrieved graph triples in *Graph-S*³ and baselines on correct answers.

**Ablation of trajectory refinement.** Removing trajectory refinement during data synthesis leads to the largest degradation among all ablations. The results indicate that without refinement, synthetic trajectories contain redundant detours, which produce noisy reward signals and undermine the stability of RL optimization.

| Retriever Train Method | Dataset | | | | | | | | | |
|---|---|---|---|---|---|---|---|---|---|---|
| | WebQSP | | CWQ | | MetaQA | | | | | |
| | | | | | 1hop | | 2hop | | 3hop | |
| | Acc | F₁ | Acc | F₁ | Acc | F₁ | Acc | F₁ | Acc | F₁ |
| *Graph-S*³ w/o step supervision | 41.83 | 53.87 | 13.47 | 16.40 | 72.63 | 81.12 | 34.97 | 45.14 | 6.43 | 11.34 |
| *Graph-S*³ | 44.29 | 58.45 | 23.62 | 30.44 | 82.77 | 92.04 | 63.17 | 76.18 | 14.70 | 36.29 |

Table 3: Performance comparison of process-level rewards and outcome-based rewards training methods.

### 4.3.2 Effectiveness of Stepwise Supervision

To further validate the effectiveness of our proposed stepwise supervision, we conducted an ablation study. Specifically, starting from the SFT-trained model, we ablated the stepwise reward signals and modified the setup to rely solely on outcome-based rewards. The results of this ablation study are shown in Table 3. Experimental results indicate that without stepwise rewards, model performance experiences a significant decline across all benchmarks, particularly on CWQ and MetaQA which involve longer reasoning chains. This confirms that fine-grained stepwise supervision enables more stable optimization and better generalization on complex multi-hop reasoning tasks.

### 4.3.3 Effective Information Quantification Analysis

To evaluate the efficiency of *Graph-S*³ in retrieving concise yet effective information, we compare it with baseline approaches by measuring the number of triples required to produce correct answers (see Figure 3). Unlike traditional methods that often retrieve large amounts of redundant information, our approach significantly reduces retrieval size while maintaining higher accuracy. In particular, our experiments show that *Graph-S*³ requires only **11.44%** of the triples retrieved by G-Retriever on average, yet still achieves superior accuracy. These results highlight the framework's ability to balance search depth with precision, thereby reducing redundancy.

## 5 Conclusion

We investigated the limitations of existing retrieval-augmented generation methods on textual graphs, highlighting their reliance on outcome-based supervision and their tendency to produce redundant or incomplete subgraphs. To overcome these challenges, we proposed a framework that integrates three key innovations: (1) A pipeline for high-quality, stepwise-supervised data synthesis; (2) Two-stage training (SFT then RL) with process-level rewards; (3) Fine-grained, interactive retrieval over textual graphs. Extensive experiments on WebQSP, CWQ, and MetaQA demonstrate that our approach consistently improves both accuracy and $F_1$, validating the effectiveness of synthetic stepwise supervision and the proposed training strategy for enhancing interactive graph retrieval.

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

## A    APPENDIX

### A.1    TRAINING DETAILS

**Implementation Details.** For data generation, we apply our proposed data synthesis pipeline, producing a total of 9,035 training instances for SFT and 3,504 instances for RL. In the SFT stage, we fine-tune the Qwen3-8B with a learning rate of $1 \times 10^{-4}$ for 3 epochs. In the RL stage, we adopt the GRPO algorithm with a batch size of $512$, 15 training epochs, a learning rate of $1 \times 10^{-5}$, a value clipping range of 0.5, and a KL divergence coefficient of 0.001. The entire RL training phase takes approximately 32 hours on 8 NVIDIA A100 80GB GPUs.

| Hyperparameter | Value |
| --- | --- |
| Learning rate | $1 \times 10^{-5}$ |
| Batch size | 512 |
| Epochs | 15 |
| Clip ratio | 0.2 |
| Gradient clipping | 1.0 |
| KL coefficient | 0.001 |
| PPO mini-batch size | 16 |

Table 4: Key hyperparameters for RL training (GRPO).

### A.2    PROMPT OF INTERACTIVE GRAPH RETRIEVAL

```
Prompts for Interactive Graph Retriever

You are an intelligent agent skilled in exploring Knowledge Graphs,
with strong reasoning abilities.  Your task is to perform question
answering over a Knowledge Graph by gradually exploring it.  You
should start from the entities mentioned in the question and explore
the graph step by step until you gather enough information to answer
the question.

Your task follows these steps:

1.  Understand the Question

2.  Analyze the Action History and Current Graph State

3.  Choose the Next Action** from the following options:

"Explore Entity":  Explore all triples directly connected to a given
entity in the Knowledge Graph.

"Choose Relation":  Select the triple(s) from the explored information
that are most relevant to the question.
```

```
Attention:  Only the triples included in the "Objects" field of the
"Choose Relation" step will be retained in the future "Current Graph
State".  So You must filter and retain the information useful for
answering the question or for further exploration.

"Finish":  Choose this action when you believe you have gathered
sufficient information to answer the question.  Your final answers
should be included in the "Objects" field.

4.  Select the Objects:  Depending on the action, provide the relevant
entity or triple(s).  Attention:  All objects must come from the
"Entities in Question" or the current "Current Graph State".  Do not
create new entities or relations.

If the action is "Explore Entity", "Objects":  ["EntityA", "EntityB"]

If the action is "Choose Relation", "Objects":  ["(Subject1,
Relation1, Object1)", "(Subject2, Relation2, Object2)"]

If the action is "Finish", "Objects":  ["Answer1", "Answer2"]

5.  Output your response in JSON format, and include a **detailed
thought process explaining your reasoning at this step.

---

Question:

Entities in Question:

Current Graph State:

Action History:

---

Please respond using the following format:

Thought Process:  <Provide a step-by-step analysis>

Action Decision:  ```json  "Action":  "<The type of action you are
taking: 'Explore Entity' | 'Choose Relation' | 'Finish'>", "Objects":
[<The entities or triples>]
```

### A.3 AN EXAMPLE OF THE REASONING PROCESS OF *Graph-S³*

```
{
    {
        "step": 0,
        "question": "the films that share directors with the film [The
            Life of Oharu] were in which genres",
        "question_entities": [
            "The Life of Oharu"
        ],
        "now_state": [],
        "action_history": [],
        "extract_res": {
            "Action": "Explore Entity",
            "Objects": [
                "The Life of Oharu"
            ]
        }
    },
    {
        "step": 1,
        "question": "the films that share directors with the film [The
            Life of Oharu] were in which genres",
        "question_entities": [
            "The Life of Oharu"
```

```
              ],
              "now_state": [
                  "(The Life of Oharu, starred_actors, Kinuyo Tanaka)",
                  "(The Life of Oharu, directed_by, Kenji Mizoguchi)",
                  "(The Life of Oharu, release_year, 1952)",
                  "(The Life of Oharu, written_by, Kenji Mizoguchi)",
                  "(The Life of Oharu, has_tags, kenji mizoguchi)",
                  "(The Life of Oharu, in_language, Japanese)"
              ],
              "action_history": [
                  "step 1, Explore Entity, Objects: ['The Life of Oharu']"
              ],
              "extract_res": {
                  "Action": "Choose Relation",
                  "Objects": [
                      "(The Life of Oharu, directed_by, Kenji Mizoguchi)"
                  ]
              }
          },
          {
              "step": 2,
              "question": "the films that share directors with the film [The
                  Life of Oharu] were in which genres",
              "question_entities": [
                  "The Life of Oharu"
              ],
              "now_state": [
                  "(The Life of Oharu, directed_by, Kenji Mizoguchi)"
              ],
              "action_history": [
                  "step 1, Explore Entity, Objects: ['The Life of Oharu']",
                  "step 2, Choose Relation, Objects: ['(The Life of Oharu,
                      directed_by, Kenji Mizoguchi)']"
              ],
              "extract_res": {
                  "Action": "Explore Entity",
                  "Objects": [
                      "Kenji Mizoguchi"
                  ]
              }
          },
          {
              "step": 3,
              "question": "the films that share directors with the film [The
                  Life of Oharu] were in which genres",
              "question_entities": [
                  "The Life of Oharu"
              ],
              "now_state": [
                  "(Ugetsu, directed_by, Kenji Mizoguchi)",
                  "(Sansho the Bailiff, directed_by, Kenji Mizoguchi)",
                  "(The Crucified Lovers, directed_by, Kenji Mizoguchi)",
                  "(The 47 Ronin, directed_by, Kenji Mizoguchi)",
                  "(The Life of Oharu, directed_by, Kenji Mizoguchi)",
                  "(Osaka Elegy, directed_by, Kenji Mizoguchi)",
                  "(The Life of Oharu, written_by, Kenji Mizoguchi)",
                  "(Sisters of the Gion, directed_by, Kenji Mizoguchi)",
                  "(The Life of Oharu, has_tags, kenji mizoguchi)",
                  "(Utamaro and His Five Women, directed_by, Kenji Mizoguchi)"
              ],
              "action_history": [
                  "step 1, Explore Entity, Objects: ['The Life of Oharu']",
                  "step 2, Choose Relation, Objects: ['(The Life of Oharu,
                      directed_by, Kenji Mizoguchi)']",
                  "step 3, Explore Entity, Objects: ['Kenji Mizoguchi']"
```

```
        ],
        "extract_res": {
            "Action": "Choose Relation",
            "Objects": [
                "(Ugetsu, directed_by, Kenji Mizoguchi)",
                "(Sansho the Bailiff, directed_by, Kenji Mizoguchi)",
                "(The Crucified Lovers, directed_by, Kenji Mizoguchi)",
                "(The 47 Ronin, directed_by, Kenji Mizoguchi)",
                "(Osaka Elegy, directed_by, Kenji Mizoguchi)",
                "(Sisters of the Gion, directed_by, Kenji Mizoguchi)",
                "(Utamaro and His Five Women, directed_by, Kenji
                    Mizoguchi)"
            ]
        }
    },
    {
        "step": 4,
        "question": "the films that share directors with the film [The
            Life of Oharu] were in which genres",
        "question_entities": [
            "The Life of Oharu"
        ],
        "now_state": [
            "(Ugetsu, directed_by, Kenji Mizoguchi)",
            "(Sansho the Bailiff, directed_by, Kenji Mizoguchi)",
            "(The Crucified Lovers, directed_by, Kenji Mizoguchi)",
            "(The 47 Ronin, directed_by, Kenji Mizoguchi)",
            "(The Life of Oharu, directed_by, Kenji Mizoguchi)",
            "(Osaka Elegy, directed_by, Kenji Mizoguchi)",
            "(Sisters of the Gion, directed_by, Kenji Mizoguchi)",
            "(Utamaro and His Five Women, directed_by, Kenji Mizoguchi)"
        ],
        "action_history": [
            "step 1, Explore Entity, Objects: ['The Life of Oharu']",
            "step 2, Choose Relation, Objects: ['(The Life of Oharu,
                directed_by, Kenji Mizoguchi)']",
            "step 3, Explore Entity, Objects: ['Kenji Mizoguchi']",
            "step 4, Choose Relation, Objects: ['(Ugetsu, directed_by,
                Kenji Mizoguchi)', '(Sansho the Bailiff, directed_by,
                Kenji Mizoguchi)', '(The Crucified Lovers, directed_by,
                Kenji Mizoguchi)', '(The 47 Ronin, directed_by, Kenji
                Mizoguchi)', '(Osaka Elegy, directed_by, Kenji Mizoguchi)
                ', '(Sisters of the Gion, directed_by, Kenji Mizoguchi)',
                 '(Utamaro and His Five Women, directed_by, Kenji
                Mizoguchi)']"
        ],
        "extract_res": {
            "Action": "Explore Entity",
            "Objects": [
                "Ugetsu",
                "Sansho the Bailiff",
                "The Crucified Lovers",
                "The 47 Ronin",
                "Osaka Elegy",
                "Sisters of the Gion",
                "Utamaro and His Five Women"
            ]
        }
    },
    {
        "step": 5,
        "question": "the films that share directors with the film [The
            Life of Oharu] were in which genres",
        "question_entities": [
            "The Life of Oharu"
```

```
        ],
        "true_answer": [
            "Drama"
        ],
        "now_state": [
            "(Ugetsu, directed_by, Kenji Mizoguchi)",
            "(Sansho the Bailiff, directed_by, Kenji Mizoguchi)",
            "(The Crucified Lovers, directed_by, Kenji Mizoguchi)",
            "(The 47 Ronin, directed_by, Kenji Mizoguchi)",
            "(The Life of Oharu, directed_by, Kenji Mizoguchi)",
            "(Osaka Elegy, directed_by, Kenji Mizoguchi)",
            "(Sisters of the Gion, directed_by, Kenji Mizoguchi)",
            "(Sisters of the Gion, has_genre, Drama)",
            "(Ugetsu, has_genre, Drama)",
            "(Ugetsu, starred_actors, Masayuki Mori)",
            "(Utamaro and His Five Women, directed_by, Kenji Mizoguchi)"
        ],
        "action_history": [
            "step 1, Explore Entity, Objects: ['The Life of Oharu']",
            "step 2, Choose Relation, Objects: ['(The Life of Oharu,
                directed_by, Kenji Mizoguchi)']",
            "step 3, Explore Entity, Objects: ['Kenji Mizoguchi']",
            "step 4, Choose Relation, Objects: ['(Ugetsu, directed_by,
                Kenji Mizoguchi)', '(Sansho the Bailiff, directed_by,
                Kenji Mizoguchi)', '(The Crucified Lovers, directed_by,
                Kenji Mizoguchi)', '(The 47 Ronin, directed_by, Kenji
                Mizoguchi)', '(Osaka Elegy, directed_by, Kenji Mizoguchi)
                ', '(Sisters of the Gion, directed_by, Kenji Mizoguchi)',
                 '(Utamaro and His Five Women, directed_by, Kenji
                Mizoguchi)']",
            "step 5, Explore Entity, Objects: ['Ugetsu', 'Sansho the
                Bailiff', 'The Crucified Lovers', 'The 47 Ronin', 'Osaka
                Elegy', 'Sisters of the Gion', 'Utamaro and His Five
                Women']"
        ],
        "extract_res": {
            "Action": "Choose Relation",
            "Objects": [
                "(Ugetsu, has_genre, Drama)",
                "(Sisters of the Gion, has_genre, Drama)"
            ]
        }
    },
    {
        "step": 13,
        "question": "the films that share directors with the film [The
            Life of Oharu] were in which genres",
        "question_entities": [
            "The Life of Oharu"
        ],
        "true_answer": [
            "Drama"
        ],
        "now_state": [
            "(Ugetsu, directed_by, Kenji Mizoguchi)",
            "(Sansho the Bailiff, directed_by, Kenji Mizoguchi)",
            "(The Crucified Lovers, directed_by, Kenji Mizoguchi)",
            "(The 47 Ronin, directed_by, Kenji Mizoguchi)",
            "(The Life of Oharu, directed_by, Kenji Mizoguchi)",
            "(Osaka Elegy, directed_by, Kenji Mizoguchi)",
            "(Sisters of the Gion, directed_by, Kenji Mizoguchi)",
            "(Sisters of the Gion, has_genre, Drama)",
            "(Ugetsu, has_genre, Drama)",
            "(Utamaro and His Five Women, directed_by, Kenji Mizoguchi)"
        ],
```

```
        "action_history": [
            "step 1, Explore Entity, Objects: ['The Life of Oharu']",
            "step 2, Choose Relation, Objects: ['(The Life of Oharu,
                directed_by, Kenji Mizoguchi)']",
            "step 3, Explore Entity, Objects: ['Kenji Mizoguchi']",
            "step 4, Choose Relation, Objects: ['(Ugetsu, directed_by,
                Kenji Mizoguchi)', '(Sansho the Bailiff, directed_by,
                Kenji Mizoguchi)', '(The Crucified Lovers, directed_by,
                Kenji Mizoguchi)', '(The 47 Ronin, directed_by, Kenji
                Mizoguchi)', '(Osaka Elegy, directed_by, Kenji Mizoguchi)
                ', '(Sisters of the Gion, directed_by, Kenji Mizoguchi)',
                 '(Utamaro and His Five Women, directed_by, Kenji
                Mizoguchi)']",
            "step 5, Explore Entity, Objects: ['Ugetsu', 'Sansho the
                Bailiff', 'The Crucified Lovers', 'The 47 Ronin', 'Osaka
                Elegy', 'Sisters of the Gion', 'Utamaro and His Five
                Women']",
            "step 6, Choose Relation, Objects: ['(Ugetsu, has_genre,
                Drama)', '(Sisters of the Gion, has_genre, Drama)']"
        ],
        "extract_res": {
            "Action": "Finish",
            "Objects": [
                "Drama"
            ]
        }
    }
}
```

## A.4 THE USE OF LARGE LANGUAGE MODELS

In this work, large language models (LLMs) were utilized solely to assist in polishing the academic writing. Specifically, the LLM (e.g., [DeepSeek]) was used to improve the fluency and clarity of the textual descriptions in Section 3 (Method) and Section 4 (Experiments). All technical content, including the core ideas, algorithms, experimental results, and analyses, was conceived and developed solely by the authors. The LLM acted strictly as a writing tool and did not contribute to the scientific reasoning or intellectual substance of the research.