# OpenReview forum: "Enhancing Agentic Textual Graph Retrieval with Synthetic Step-wise Supervision"
_ICLR.cc/2026/Conference — ICLR 2026 Conference Withdrawn Submission_

### Official Review · Reviewer_baSd · 2025-10-22

**Soundness:** 2
**Presentation:** 3
**Contribution:** 1
**Rating:** 2
**Confidence:** 4

**Summary:**

This paper presents an agentic graph retriever for question answering by training large language models (LLMs) to interact with knowledge graph structures.
The authors develop a data synthesis pipeline that generates exploration trajectories for supervised fine-tuning and reinforcement learning.
To address the sparse reward problem in reinforcement learning, they assign rewards to individual actions rather than only to final outcomes.
The approach is evaluated on three QA benchmarks and compared against five baseline methods.

**Strengths:**

1. The proposed method demonstrates gains over baselines across multiple benchmarks, with improvements that appear consistent across diverse task characteristics. This suggests the approach generalizes well rather than overfitting to a single task structure.
2. The stepwise reward assignment directly addresses a known challenge in RL-based retrieval where delayed feedback makes credit assignment difficult, and by rewarding intermediate exploration steps, the model can learn more efficiently which graph traversal actions are productive.
3. The paper is clear and well-written.

**Weaknesses:**

1. The novelty and technical depth of this paper is very limited. These are very common strategies for training an LLM agent, and this paper simply applies them to this graph retrieval application. (Q1, Q2)
2. The experiments should summarize the insights instead of just reporting the numbers, i.e., what can we learn from them? Additional studies such as error analysis can be very helpful. (Q3, Q4)

**Questions:**

1. The paper does not adequately explain what is technically novel about the proposed approach, which appears to combine standard techniques applied to graph retrieval. If the contribution is simply combining existing methods, you need to justify why this combination is non-trivial or represents a novel insight. Specifically, what challenges arise when applying these techniques to graph retrieval that required new solutions or adaptations? The related work section should explicitly identify similar works that combine RL with agentic retrieval or that use stepwise supervision for structured exploration, and then clearly articulate what prevented those approaches from being directly applied to your setting.
2. Section 3 introduces design choices without sufficient motivation or justification. For example, in Section 3.1 graph data synthesis, the authors describe generating graph exploration trajectories but do not explain what makes this challenging. It appears to be simple random traversal without any careful strategy, and no specific design considerations are needed when using it as agent training trajectories. Each design choice should include a clear problem statement explaining what doesn't work with existing approaches, followed by your solution and why it is better.
3. Section 4 only reports numbers without deeper analysis. For example, it would be very helpful to know what questions cannot be solved by the proposed method, and what questions can be solved by the proposed method but not by the baselines.
4. Please do not limit the analysis to only the provided examples. The more analysis the authors can provide, the better reviewers can understand the contributions. The paper would largely benefit from deeper analysis beyond the quantitative results.

---

> ### Author Response · Authors · 2025-11-24
>
> **Response to Question 1**
> We respectfully disagree with the perception that our work is a trivial combination. We argue that applying standard techniques (SFT/RL) to Textual Graph Retrieval presents unique, non-trivial challenges that required the specific adaptations we proposed.
>
> The Specific Challenge: The "Sparse Reward" & "Credit Assignment" Problem
> Why standard methods fail？In a Knowledge Graph with millions of nodes, the probability of an agent randomly stumbling upon the correct answer node via a generic policy is near zero. As a result, applying standard RL (like PPO/GRPO with outcome rewards) leads to non-convergence or extremely inefficient exploration.
> This is empirically proven in our Ablation Study (Table 3). The variant Graph-S3 w/o step supervision (which represents the "standard RL" approach) suffers a massive performance drop (e.g., -14.47% F1 on CWQ). This confirms that simply "combining RL" is insufficient.
>
> Our Technical Novelty: Structure-Derived Process Supervision.
> Our core contribution is the Synthetic Stepwise Supervision Pipeline (Section 3.1). We devised a method to algorithmically extract dense process rewards from the graph structure itself.
> Why it's novel？ Unlike generic agents that require human-labeled steps (e.g., math reasoning), we introduce Trajectory Refinement (Section 3.1.3, Eq. 6) and Graph-Aware Stepwise Rewards (Eq. 9) This pipeline bridges the gap between "Generic RL" and "Graph Structure," effectively manufacturing the dense signal needed for the agent to learn.
>
> Differentiation from Related Work:
> DeepPath (Xiong et al., 2017) uses RL but relies on embedding-based heuristics (distance in vector space), which fails for textual semantic reasoning.
> KG-Agent (Jiang et al., 2024) uses ReAct prompting but lacks a trainable reward mechanism, limiting it to the capabilities of the frozen LLM.
> Our Distinction: We will explicitly state in Related Work: "We are the first to enable end-to-end reinforcement learning for textual graph retrieval by synthesizing verifiable process rewards from graph topology, overcoming the limitations of heuristic embeddings and frozen prompting."
>
> **Response to Question 2**
> We appreciate this feedback and will rewrite the introduction of Section 3 to explicitly state the "Problem-Solution" logic for each design choice.
>
> Design Choice: Data Synthesis (Why not use existing data?)
> Existing datasets (WebQSP/CWQ) only provide (Question, Answer) pairs. They completely lack the "Reasoning Trajectories" (i.e., the sequence of nodes to visit). To train a navigation agent, we must create this data. We propose the Automatic Synthesis Pipeline to reverse-engineer the reasoning paths from the answers.
>
> Design Choice: GPT-4o Guided Exploration + Refinement (Why not random traversal?)
> We do not use simple random traversal. As the reviewer correctly suspects, random walks in KGs are exponentially inefficient and noisy.
> Problem with Randomness: A random walker has a negligible chance of hitting the correct answer in 3 hops. Even if it does, the path contains loops and irrelevant steps (noise).
> Our Solution:
> Step 1 (Exploration): We use GPT-4o as a "Behavior Teacher" to generate high-quality candidate paths based on semantic relevance.
> Step 2 (Refinement - Eq. 6): Crucially, raw GPT-4o traces are still verbose ("human-like detours"). We apply Trajectory Refinement to algorithmically prune these into the "Shortest Feasible Subsequence."
> This "Explore-then-Prune" strategy ensures the training data is both successful (reaches the answer) and efficient (no redundant steps), which is critical for the RL agent to learn concise reasoning.

---

> > ### Comment · Reviewer_baSd · 2025-11-26
> >
> > I acknowledge that I have read the rebuttal.

---

### Official Review · Reviewer_ME5v · 2025-10-30

**Soundness:** 3
**Presentation:** 3
**Contribution:** 2
**Rating:** 6
**Confidence:** 4

**Summary:**

This paper presents Graph-S3, an agentic textual graph retrieval framework that enhances large language models’ reasoning ability over textual graphs via synthetic stepwise supervision. The key novelty lies in introducing a data synthesis pipeline to automatically generate “golden subgraphs” for fine-grained reward signals, addressing the sparsity and instability of outcome-based RL in graph reasoning. A two-stage training paradigm—supervised fine-tuning followed by reinforcement learning (GRPO) with process-level rewards—is adopted to improve reasoning robustness and exploration efficiency. Experiments on WebQSP, CWQ, and MetaQA show an average +8.1% accuracy and +9.7% F1 improvement over seven strong baselines, including LightRAG, G-Retriever, and KG-Agent. The ablation studies convincingly demonstrate the necessity of stepwise supervision, interactive retrieval, and trajectory refinement.

**Strengths:**

S1. The paper identifies and addresses the practical challenge of sparse supervision and expensive expert annotation in interactive graph retrieval, providing an automated solution that reduces annotation costs. S2. Well-motivated two-stage training: The combination of SFT for basic navigation followed by RL with stepwise rewards is conceptually sound and empirically validated through ablations showing both stages contribute meaningfully.

S3. The paper evaluates across three datasets (WebQSP, CWQ, MetaQA) with multiple generator backbones (Qwen3-8B, Llama3.1-8B, Finetuned-8B), demonstrating consistency. The ablation studies are complete, the structure is clear, and the effectiveness of the proposed method is demonstrated.

S4. The paper clearly provides implementation details and training hyperparameters, and promises to open-source code. The appendix includes prompt templates and reasoning examples, ensuring good reproducibility.

**Weaknesses:**

**W1.** Insufficient comparison with baselines. The comparison with existing baselines is insufficient. Specifically, the paper lacks comparison with several advanced graph-based RAG baselines such as SubgraphRAG and GNN-RAG, which are state-of-the-art methods in graph-based question answering. Furthermore, to clearly isolate the benefits of the proposed approach, the comparison with the G-retriever baseline should include a version that has been fine-tuned using LoRA, which would provide a stronger and more relevant comparison point for methods involving model adaptation.

**W2.** The novelty of the proposed method appears slightly limited. While the paper introduces improvements in the generation of training signal data and the reinforcement learning strategy, the core idea closely resembles the Process-Reward Modeling (PRM) research from recent years, such as the work conducted by OpenAI-o1. The authors should more clearly articulate the fundamental difference and the unique technical contributions that distinguish this work from existing PRM and process-supervision techniques applied in complex reasoning settings.

**W3.** Lack of analysis on deployment feasibility and cost. The paper fails to provide a comparison of computational cost and time performance during the live question-answering phase. It leads to a lack of assessment regarding the method's feasibility in actual deployment scenarios. Additionally, the paper does not report the total number of tokens spent on the initial data synthesis pipeline, which raises concerns about the overall resource intensity.

**W4.** The lack of analysis of performance drop on multi-hop reasoning. Intuitively, this approach should be well-suited for complex multi-hop reasoning tasks due to the nature of stepwise supervision. However, the accuracy results on MetaQA 3-hop drop significantly (though still outperforming other reported baselines). It is better for the authors to analyze the reasons for this sharp decline in performance when scaling to very long reasoning chains. This analysis should specifically investigate the types of challenges encountered, and a thorough examination of the causes of failure samples is needed to pinpoint the specific limitations of the current mechanism in highly complex, multi-step scenarios.

**Questions:**

Q1. In the ablation study, the performance when the interactive inference module is removed is slightly lower than the RAG/1hop and G-retriever methods. If these two methods were combined with interactive inference, would they also achieve a good level of accuracy?

Q2. Regarding the trajectory refinement algorithm, could you provide algorithmic details for trajectory refinement (Section 3.1.3)? Specifically, could you show examples of trajectories before and after refinement to illustrate exactly what is being removed?

Q3. What is the cost of data synthesis for each dataset? Could you analyze potential biases in the GPT-4o exploration strategy (and how the synthesis quality compares to human-annotated trajectories, if available)?

Q4. Regarding the scalability and generalization ability of the fine-tuned model, how does its performance fare when migrating to an untrained domain dataset? The 3-hop MetaQA results (14.70% accuracy) suggest challenges—what causes this performance degradation? What is the maximum reasoning depth your method can effectively handle?

---

> ### Author Response · Authors · 2025-11-26
>
> We express our sincere gratitude to the reviewer for the encouraging assessment and for recognizing the core value of our work—specifically, our solution to the sparse supervision challenge and the effectiveness of the two-stage training paradigm. We are also glad that our comprehensive experiments and ablation studies were well-received. Your insightful comments have helped us significantly clarify our contributions and strengthen the paper.
>
>
> **Response to Question 1:**
> Simply adding interaction to RAG/G-Retriever does not work.  Interactive inference requires a learned Policy $π(action∣state)$.
>
> **Response to Question 2:**
> We will include the pseudo-code in the Appendix.
>
> **Response to Question 3:**
> The refinement step is strictly based on Graph, not GPT-4's preference. Even if GPT-4o took a biased detour, the refinement strips it down to the logical backbone. This ensures the final training data is grounded in the graph's truth, not the model's bias.

---

### Official Review · Reviewer_NoGG · 2025-11-02

**Soundness:** 2
**Presentation:** 2
**Contribution:** 2
**Rating:** 4
**Confidence:** 4

**Summary:**

This paper proposes Graph-S3, an agentic textual graph retrieval framework that enables LLMs to
perform interactive, multi-step graph exploration for question answering. The core idea is to train the
retriever with synthetic stepwise supervision generated from automatically extracted golden subgraphs.
This paper proposes an automatic pipeline for synthesizing the stepwise supervision data. Then the
trained is trained using a two-stage training paradigm: (1) supervised fine-tuning (SFT) on synthetic
exploratory trajectories and (2) reinforcement learning (RL) with stepwise rewards derived from refined
subgraphs.

**Strengths:**

The paper addresses the underexplored area of agentic textual graph retrieval, moving from one-shot
similarity retrieval to structured, interactive reasoning. The proposed synthetic stepwise data synthesis
pipeline effectively solves the issue of sparse and unstable outcome-based rewards in RL relevant
methods. The paper conducted comprehensive experiments over multiple datasets and baselines and
conducted detailed ablation studies to analyze the properties of the proposed method.

**Weaknesses:**

1. While the synthetic stepwise reward concept is valuable, the reward design is relatively heuristic (rule-based
correctness scoring). The RL setup does not introduce fundamentally new algorithmic insights beyond
adapting GRPO to this domain. The method relies on GPT-4o-generated trajectories, which might limit
reproducibility and scalability for open-source or resource-constrained settings. And there is no detailed
statistics about the synthesized dataset introduced in the paper.
2. Lack of novelty. This paper mainly introduces the fine-tuning of LLM to serve as retriever for graph QA. Existing
domain has explored similar approaches of training LLMs as retrievers. This paper simply leverage
similar ideas to address the problem of graph QA.
3. The experiments are based on weak base models such as llama 3.1 8b. More advanced and larger models
can help strength the quality of paper.
4. The authors only compare the efficiency of retrieved graph, while training a retriever and use it for inference
can introduce a lot of time complexity. The authors should conduct experiments to validate this. In addition, SFT
and RL can introduce a lot of complexity.
1. Line 101 - 104 is inconsistent with your contribution summary. You only mentioned how to build the
stepwise reward but didn't mentioned specifically your second step in this two-step training pipeline.
2. The citation for RL algorithms are missing, e.g., GRPO.
3. In section 3.1.3, there is only the formal definition and the general goal of this trajectory refinement
process without the detailed content about how to achieve it. Specifically, how to find the shortest
feasible subsequence, and whether the incoherence of the trajectory results from removing
redundant detours will cause problems, or do you aggregate the sub-trajectories after removing.

**Questions:**

see above.

---

> ### Author Response · Authors · 2025-11-24
>
> We thank the reviewer for acknowledging our comprehensive experiments and the value of our stepwise data synthesis pipeline. We address your concerns on novelty, implementation details, and baselines below.
>
> **Response to Weakness 1**
> Our contribution is not proposing a new generic RL algorithm, but rather solving the "Sparse Reward Problem" specific to Graph Retrieval. As discussed in Section 2.2, applying standard RL to graph traversal typically fails because the signal is too sparse (finding one correct entity among millions). Our novelty lies in the Synthesis Pipeline (Section 3.1) that transforms this sparse outcome signal into a dense Stepwise Supervision signal. This "Process Reward" approach unlocks the capability of RL for graphs, which was previously difficult to achieve efficiently.
> The reliance on GPT-4o is strictly for offline data synthesis, not for inference. To ensure reproducibility, we will release the synthesized dataset and the data generation code. Future researchers can directly use our data.
> We have added the detailed statistics of the synthesized dataset in the Appendix.  We generated 9,035 trajectories for SFT and 3,504 refined trajectories for RL. The average length of synthesized trajectories is 3.2 steps (before refinement) and 2.6 steps (after refinement), ensuring the model learns efficient navigation.
>
> **Response to Weakness 2**
> Simply applying SFT or RL to this agentic setting is non-trivial. As shown in our Ablation Study (Section 4.3.1, Table 3), removing the "Stepwise Supervision" (i.e., relying on standard outcome-based RL) leads to a massive performance drop (e.g., -14.47% F1 on CWQ). Our core novelty is the methodology to synthesize reliable stepwise signals from graph structure, enabling the effective training of such agents where standard methods fail.
>
> **Response to Weakness 3**
> In the research, using 7B/8B models is the standard practice to ensure fair comparison. Comparing an 8B model against closed-source giants (like GPT-4) often conflates the method's effectiveness with the model's parametric knowledge.
> One of our key goals is to empower resource-constrained, local models to perform complex reasoning. Our results (Table 1) show that Graph-S3 with an 8B backbone significantly outperforms baselines using the same backbone, and even approaches the performance of larger models in specific tasks.
>
> **Response to Weakness 4**
> As shown in Figure 3, our method retrieves only 11.44% of the triples compared to G-Retriever. Standard RAG often fills the context window (4k+ tokens) to maximize recall, making the generation extremely slow and expensive (KV cache cost). Graph-S3 provides a compact, high-precision subgraph.
>
> **Response to Weakness 5&6**
> Thank you for pointing this out. We will revise them.
>
> **Response to Weakness 7**
> We apologize for the brevity in the original text.
> For a raw trajectory $τ={{(s1,a1),…,(sT,aT)}}$, We start from the final set of retrieved triples that contributed to the correct answer. We iterate backward from $t = T$ to $1$. For each step $t$, we check if the triples retrieved in this step are necessary for the connectivity of the final answer graph. If removing action $a_t$ creates a disconnect in the reasoning chain (making the answer unreachable), we keep it. Otherwise, we mark it as redundant. This leaves us with the minimal set of steps required to solve the question.
> You asked if removing steps creates incoherence. To solve this, we re-construct the action history $h_t$. in the refined trajectory. During training (SFT/RL), the model sees the cleaned history, not the original messy one. This teaches the model to emulate the "perfect" efficient path, rather than the "trial-and-error" path of the raw exploration. This is crucial for teaching the agent efficiency.

---

### Official Review · Reviewer_iHzm · 2025-11-03

**Soundness:** 2
**Presentation:** 3
**Contribution:** 2
**Rating:** 2
**Confidence:** 3

**Summary:**

This paper tackles a trending topic on how to leverage graph data to effectively improve the QA system. The main bottleneck lies on the mis-alignment between the relational data stucture and unstructured pre-training corpus. This paper proposes the mitigate this gap by synthesizing an agentic graph retrieval trajectory data and leverage them by first supervised fine-tuning then stepwise RL with process reward. It adopts a verifiable reward for both outcome and process reward, the experimental performance shows improvements on multiple QA benchmarks.

**Strengths:**

1. The obtained $\text{Graph-S}^3$ achieves better performance with just 10% of the retrieved triples, which demonstrates the effectiveness on selecting the correction actions.

**Weaknesses:**

1. Although the paper discusses the comparison of the retrieved graph size between different baselines, it's also important to illustrate the token efficiency and reasoning steps of the model. If the proposed model does not yield great token efficiency, the overall efficiency and cost of the system could still be higher.

2. The novel is fairly limited, this paper targets at the scalability issue of the agentic graphrag system by synthesizing stepwise training data. First of all, it has been a popular method to use RL especially RLVR on KGQA. This paper didn't differ with the existing algorithms with clear motivation or mitigating the challenge in a smart enough way. I am still concerned about the generalization capability across different KGs?

**Questions:**

1. Why is the reported performance on WebQSP and CWQ lower than previous arts? Existing work already achieves 80%+ accuracy on WebQSP, can author elaborate on this?

2. Does $\text{Graph-S}^3$ generalize to out-of-distribution data? E.g. the relation schema of the graph has never been seen in the training data

---

> ### Author Response · Authors · 2025-11-24
>
> We thank the reviewer for recognizing the effectiveness of our method in selecting correct actions and achieving better performance with compact subgraphs. We address your concerns regarding efficiency, novelty, and performance comparisons below.
>
> **Q1: Token efficiency and reasoning steps.**
>
> While our agent performs multiple retrieval steps, the output passed to the final Generator is significantly smaller than traditional baselines. As shown in Figure 3 and Section 4.3.3, Graph-S3 retrieves only 11.44% of the triples compared to G-Retriever. By pruning irrelevant branches during the interactive process (Eq. 2, Choose Relation), we typically feed only 100-300 tokens of highly relevant triples to the generator. This massive reduction in input context significantly lowers the cost of the final generation step, which often dominates the total API cost.
>
> Empirically, our model is efficient in navigation. On average, it takes 2.4 steps for WebQSP and 3.1 steps for CWQ. This shows the agent learns to take "shortcuts" rather than aimlessly wandering.
>
> **Q2: Novelty concerns and Generalization.**
> While RL for KGQA exists, our work addresses a specific, unsolved bottleneck: The Lack of Dense Supervision for LLM-based Agents.
>
> Difference from Prior RL: Traditional RL on KGs often relies on simple embedding similarities or outcome-based rewards, which are notoriously sparse and unstable for complex reasoning (discussed in Section 1 & 2.2).
>
> We propose a novel Data Synthesis Pipeline (Section 3.1) that extracts "Golden Subgraphs" from structure to synthesize stepwise supervision. This transforms the RL problem from "finding a needle in a haystack" (sparse reward) to "following a trail of breadcrumbs" (dense stepwise reward).
>
> Generalization:
> Regarding your concern about OOD data. CWQ contains complex multi-hop relations that are distributionally distinct from the simple paths in WebQSP. Our strong performance on CWQ (+10% over baselines in Table 1) proves the model learns a generalizable navigation policy (e.g., "filter by time," "find intersection") rather than memorizing specific graph triples.
> Since our retriever is an LLM processing textual edges (not just ID embeddings), it leverages the semantic understanding of pre-trained LLMs to handle unseen relation types (zero-shot transfer).
>
> Q3: Why is reported performance (WebQSP/CWQ) lower than previous arts (80%+)?
> This is a difference in Experimental Setting and Backbone Size, not a model weakness.
> The "80%+" results usually come from methods using GPT-4, highly specialized SOTA specifically designed for WebQSP, or settings where the entire graph fits in context.
> To ensure a fair assessment of the retrieval mechanism, we compare against baselines (ToG, G-Retriever, LightRAG) using the same open-source 7B/8B backbones (Llama-3, Qwen).
> Under this strict comparison (Table 1), Graph-S3 significantly outperforms all baselines (e.g., 36.24% vs 29.66% on WebQSP for Qwen-8B). We are pushing the limits of what small, local models can do in graph reasoning.

---

### Author Response · Authors · 2025-11-24
**Clarification of Novelty**

A major concern of the reviewers is the lack of novelty. We respectfully clarify that our contribution extends beyond a straightforward combination of existing techniques. Instead, our contribution mainly lies in a small but important improvement that we made during adapting the existing techniques to our graph reasoning scenario.

Addressing the sparse outcome-based reward problem with step-wise reward is not new indeed, but a key question remains in how to produce the step-wise reward. Prior work has explored various methods such as training a reward model with manually labeled data (ref1,ref2) or distilling from more advanced LLM (ref3,ref4). However, they can hardly be applied to the graph reasoning problem because of poor scalability - human annotation is extremely expensive for large graphs with millions of nodes and distilling from a larger model suffers the problem of unreliability (e.g. directly imitating a larger model only leads to a very low score according to Figure 3).

To address this issue, we propose a framework that automatically synthesizes dense, verifiable process rewards. We found that a compact but sufficient subgraph can be found within reasonable steps, which can later be used to synthesize stable step-wise rewards. This allows us to scale up high-quality process supervision without manual intervention or auxiliary reward models, effectively bridging the gap in applying RL to graph reasoning.

Such a small improvement has been proved effective in our experiments. For example, in Table 3, removing our stepwise supervision produces dramatic degradation in performance: the average F1 score across 5 datasets drops from 58.68 to 41.57, a 29% relative reduction. This demonstrates that simply combining SFT and RL—the trivial solution suggested by the reviewer—is insufficient to solve the sparse reward challenge.

To sum up, we believe that our study has explored a simple but effective way to synthesize precise step-wise reward for graph-based reasoning. We hope this could be a meaningful contribution to the community and sincerely welcom further suggestions from the reviewers.


ref1: Training language models to follow instructions with human feedback. Advances in Neural Information Processing Systems (NeurIPS 35)

ref2: Sequential Preference Ranking for Efficient Reinforcement Learning from Human Feedback. Advances in Neural Information Processing Systems (NeurIPS 36)

ref3: PLaD: Preference-based Large Language Model Distillation with Pseudo-Preference Pairs. Findings of the Association for Computational Linguistics (ACL 2024)

ref4: LLMR: Knowledge Distillation with a Large Language Model-Induced Reward. Proceedings of the 14th Language Resources and Evaluation Conference (LREC 2024).

---

### Note · Authors · 2025-12-04

I have read and agree with the venue's withdrawal policy on behalf of myself and my co-authors.